# Bridging Gaps with Dynamic Knowledge Probes: Robust LLM–KG Collaborative Reasoning

## Abstract

Large Language Models (LLMs) exhibit exceptional capabilities in various natural language tasks but are constrained by static knowledge, potential hallucinations, and opaque reasoning processes. Integrating external Knowledge Graphs (KGs) has emerged as a promising solution. While agent-based paradigms enhance knowledge exploration by iteratively retrieving grounded facts from KGs, they often adopt a conservative KG-centric strategy that deliberately avoids using the LLM's internal knowledge—rendering them vulnerable to failures whenever missing links occur, a common challenge even in largely complete KGs. We propose a KG–LLM collaborative framework that repositions the LLM's knowledge as dynamic knowledge probes, generated via our Guidance Graph of Thought (GGoT) reasoning backbone from partially specified triples. These probes guide KG exploration, highlight potential incompleteness, and trigger trust-aware bridging with existence and necessity checks before integrating LLM-derived entities. Cross-triple constraint-based disambiguation then ensures consistency, using KG structure for credible nodes and LLM validation for low-confidence ones. Extensive experiments across multiple benchmarks show that our framework consistently achieves superior performance over existing approaches, with ablation studies verifying the contribution and necessity of each component in our design.

## 1 Introduction

Large Language Models (LLMs) have demonstrated remarkable performance across a wide range of natural language processing tasks, including question answering Wang et al. (2024b); Li et al. (2024); Zhao et al. (2024)and commonsense reasoning. By leveraging deep architectures pre-trained on massive corpora, they are capable of generating coherent and contextually appropriate responses Ji et al. (2024); Chen et al. (2023; 2022); Gong & Sun (2024). However, their knowledge remains frozen after training, and they are prone to producing plausible yet factually incorrect outputs—a phenomenon known as hallucination Bang et al. (2023); Ji et al. (2023); Luo et al. (2023b). Knowledge Graphs (KGs), on the other hand, offer structured, explicit, and verifiable representations of entities and their relationships, making them well-suited to compensate for these limitations of LLMs Zhang et al. (2019b); Yao et al. (2019); Wang et al. (2021); Luo et al. (2023a).

Existing integration strategies can be broadly categorized into two types. Simple coupling methods enable the LLM to directly generate KG queries and retrieve relevant facts on demand. While straightforward, these approaches treat the KG as a black box and may fail when the generated queries do not align with the schema or content of the graph. Agent-based exploration methods—such as ToG Sun et al. (2024a), PoG Chen et al. (2024), FiSKE Tao et al. (2025b), and GG-explore Tao et al. (2025a)—allow the LLM to iteratively explore subgraphs of the KG. These paradigms enhance reasoning robustness by decomposing complex queries and using the KG for grounded fact retrieval.

However, these agent-based paradigms remain fundamentally KG-authoritative and notoriously brittle: they strictly rely on the KG as the sole source of truth to avoid hallucinations, but this rigidity becomes their Achilles' heel. The entire reasoning chain can catastrophically fail even if a single link is missing in the KG. This all-or-nothing reliance on KG completeness is a critical limitation,

Figure 1: Illustration of KG Reasoning under Incompleteness: Existing Approaches vs Trust-aware Bridging in Our Framework.

as real-world KGs are inherently incomplete. A truly robust framework must not only leverage the KG's verifiability but also compensate for its incompleteness, as illustrated in Fig. 1.

To overcome this brittleness, we propose a new explicit LLM–KG collaborative exploration framework that tightly couples the complementary strengths of both components. ~~The LLM~~ In our design, the LLM is not used as an unconditional source of truth; rather, it supplies probes—lightweight, LLM-generated candidate entities treated as hypotheses that suggest where to look in the KG—to (i) steer exploration toward semantically relevant subgraphs, (ii) flag points where the KG may be incomplete, and (iii) when needed, trigger a trust-aware bridging~~.~~ process, in which LLM suggestions are cross-checked before incorporation.

Our framework is built on our proposed Guidance Graph of Thought (GGoT), an LLM-only multi-step reasoning backbone. Rather than producing one-shot answers, GGoT progressively elicits the model's parametric knowledge across steps, classifies triples by specificity, and iteratively converts partially specified triples (Type II) into fully specific triples (Type I) via knowledge mining and constraint disambiguation. Crucially, GGoT is decoupled and thus highly extensible: each step exposes a clean interface that outputs a vetted set of facts; subsequent steps consume only these facts—not prior textual rationales. This modular interface is the extension point that lets us interleave KG verification and bridging between steps and scale GGoT from an LLM-only process to a tightly integrated LLM–KG co-exploration without changing step logic. We extend GGoT with three key mechanisms: (1) **Guided KG Exploration via LLM Knowledge Probes**, where LLM-generated candidate entities are intersected with KG candidates, expanded into high-confidence sets, and verified via semantic consistency filtering to narrow the search space; (2) **Trust-aware Bridging of KG Knowledge Gaps**, which performs existence and necessity checks before temporarily integrating LLM-suggested entities, and removes them if they fail to connect to credible knowledge in later steps; (3) **Constraint-Based Disambiguation in KG-LLM Collaboration**, enforcing consistency across triples sharing a generic entity by using KG structure for credible nodes and LLM cross-validation for low-confidence bridge nodes.

In summary, our main contributions are as follows:

- We propose the **Guidance Graph of Thought (GGoT)** framework — an LLM-only, multi-step reasoning backbone that classifies triples by specificity and iteratively converts partially specified triples into fully specific ones via knowledge mining and constraint-based disambiguation. Its decoupled, fact-level interface between steps enables clean integration points for external modules.

- Building on GGoT, we design a **KG–LLM collaborative reasoning extension** that interleaves KG verification and LLM reasoning. This extension introduces three mecha-

nisms: (i) *Guided KG Exploration via LLM Knowledge Probes* to direct exploration toward semantically relevant subgraphs and narrow the candidate scope; (ii) *Trust-aware Bridging of KG Knowledge Gaps* with existence and necessity checks before safely integrating LLM-suggested entities; and (iii) *Constraint-based Disambiguation* to enforce cross-triple consistency using KG structure for credible nodes and LLM cross-validation for low-confidence ones.

- Experimental results demonstrate that our method consistently outperforms existing approaches across diverse datasets and remains highly effective even for LLMs with fewer parameters, highlighting its practical utility. Ablation studies further validate the effectiveness of each module, particularly the designed trustworthy knowledge probe. This work offers a new perspective for reliably leveraging internal knowledge in LLMs.

## 2 RELATED WORK

**Reasoning with Large Language Models.** Recent work has improved LLM reasoning via structured prompting. DecomP He et al. (2021) decomposes tasks into sub-tasks. Chain-of-Thought (CoT) Wei et al. (2022a) and variants—Tree-of-Thought (ToT) Yao et al. (2023), Graph-of-Thought (GoT) Besta et al. (2024), Memory-of-Thought (MoT) Li & Qiu (2023)—generate intermediate reasoning steps. Plan-and-Solve prompting Wang et al. (2024a) guides LLMs to formulate and execute plans.

**Integrating Knowledge Graphs with LLMs.** Knowledge Graphs (KGs) provide structured knowledge to support reasoning. Early methods embed KG knowledge during pre-training or fine-tuning Zhang et al. (2019a), while retrieval-augmented approaches Li et al. (2023b); Linders & Tomczak (2025); Baek et al. (2023) dynamically fetch relevant facts. Agent-based paradigms such as ToG Sun et al. (2024b); Xu et al. (2024) and UniKGQA Jiang et al. (2022); Zhao et al. (2022) traverse KGs iteratively to answer complex queries. Although effective for sparse knowledge graphs, many-to-few strategies He et al. (2024) pose scalability and efficiency challenges in dense KG settings. Surveys and benchmarks have analyzed KGQA methods Cohen et al. (2023); Sen et al. (2021); Saffari et al. (2021).

## 3 GUIDANCE GRAPH OF THOUGHT

We first introduce the **Guidance Graph of Thought (GGoT)**, which serves as the core representation in our framework. It organizes the input query into a structured form, enabling explicit decomposition of reasoning steps and subsequent KG-LLM collaborative mechanisms. This section presents the construction of GGoT, the categorization of its triples, and the conversion process from generic to specific entities.

### 3.1 GUIDANCE GRAPH CONSTRUCTION

Natural language queries often mix concrete entities (*e.g.,* "Beijing") with abstract concepts or semantic types (*e.g.,* "city"). Direct reasoning over raw text makes it difficult to control and verify logical steps. Encoding these queries as a structured graph separates known facts from reasoning targets, allowing precise manipulation in downstream modules.

Following the structured rules and procedures of Tao et al.Tao et al. (2025a), we convert each query into a declarative sentence and identify specific entities (concrete entities, e.g., 'Beijing') and generic entities (semantic types, e.g., 'city'). We then integrate these elements into a **structured graph** where nodes are specific or generic entities and edges are labeled relations extracted from the query. This Guidance Graph forms the foundational input to our method. The detailed construction rules are in Appendix A.3. Prompt templates used for construction are presented in Tables A1 and A2.

### 3.2 SPECIFICITY-DRIVEN ITERATIVE TRIPLE RESOLUTION

Triples in the Guidance Graph $(h, r, t)$ vary in their specificity: some encode fully grounded facts, while others contain generic placeholders requiring inference. This variability calls for a systematic

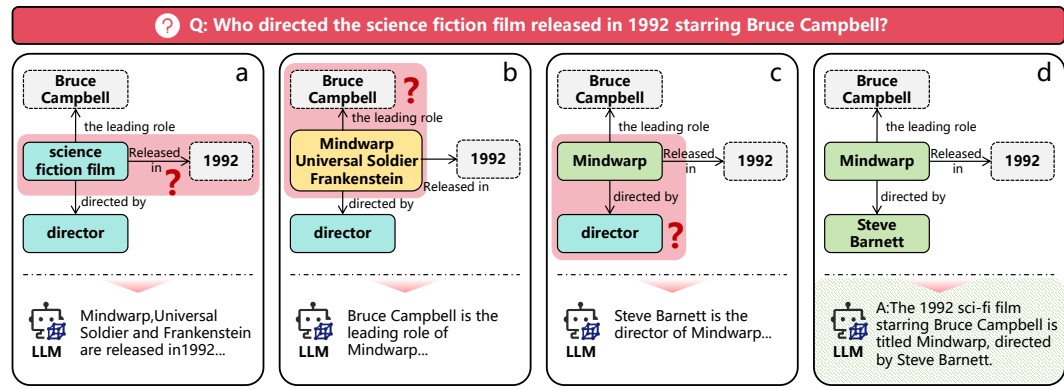

Figure 2: Specificity-driven Iterative Resolution of the Guidance Graph: An Example.

process to resolve triples — turning ambiguous generic entities into specific ones — in an iterative manner. By progressively reducing uncertainty, we enable coherent and verifiable reasoning over the graph.

We define three mutually exclusive categories:

- **Type I**: Both head $h$ and tail $t$ are specific entities (China, Capital, Beijing). These carry explicit facts and often emerge through reasoning from less specific triples.

- **Type II**: One side specific, the other generic (China, Capital, City). Ambiguous information in such triples can be resolved using the specific entity as a clue.

- **Type III**: Both head and tail are generic (Country, Capital, City), containing no concrete facts and requiring dependency on other triples to resolve.

**Iterative Resolution Process**   Resolution begins with Type II triples. We invoke the internal knowledge of large LLMs to resolve the generic entity into concrete candidates, producing new Type I triples. When a generic entity in a Type III triple becomes specific via connections to resolved triples, the triple is promoted to Type II. This iterative reduction process continues until all triples are fully specific (Type I), marking reasoning completion. Figure 2 illustrates the complete iterative resolution process of the guidance graph through an example.

### 3.3   Iterative Triple Conversion

Triple conversion is the core operation within the iterative resolution process. It specifies how a Type II triple is transformed into a Type I triple through knowledge mining and constraint-based disambiguation.

**Knowledge Mining:** For a given Type II triple $(s, r, g)$, where $s$ is a specific entity, $r$ is a relation, and $g$ is a generic entity, we treat $s$ and $r$ as reasoning clues and $g$ as the inference target. We prompt the LLM to mine concrete knowledge about this target from its internal parametric knowledge (see the Appendix A3 for detailed prompts). This process generates one or more candidate specific entities for the generic entity $g$, effectively transforming the original Type II triple into Type I triple.

**Disambiguation:** In a knowledge graph, a single generic entity $g$ often participates in multiple Type II triples, each connected to a different specific entity $s_i$. These interconnected $s_i$ entities collectively form multi-faceted constraints on $g$. Therefore, after mining a set of candidate values for $g$ from one triple (e.g., $(s_1, r_1, g)$), we must leverage these pre-existing constraints from other related triples for disambiguation. As illustrated in Step 2 of the figure, we use a specific entity $s_2$ and its relation $r_2$ from another connected triple $(s_2, r_2, g)$ as new clues. We prompt the LLM to verify and filter the candidate set, specifically by eliminating candidates that conflict with the context or commonsense knowledge implied by $s_2$ and $r_2$. This step enhances the reliability of the final resolved entity. Detailed prompts are provided in the Appendix A4.

# 4 KNOWLEDGE GRAPH–AUGMENTED TRIPLE CONVERSION VIA LLM PROBES

To mitigate the hallucination issue of large language models and enhance the reliability of knowledge reasoning, we extend the LLM-only GGoT approach to a collaborative framework into a collaborative paradigm that integrates both knowledge graphs and LLMs. GGoT is originally designed for triple conversion as a decoupled process, where each step depends solely on the output of its predecessor, independent of the intermediate reasoning process. Leveraging this decoupling property, we incorporate verified knowledge from a KG to augment and restructure triple conversion within the GGoT framework. In this augmented setting, KG-verified facts serve as the primary trusted foundation, ensuring the reliability of converted triples and the inferred relational paths. The internal parametric knowledge of the LLM is employed strategically as a *knowledge probe* in this process, serving as a complementary guide rather than the primary source of facts. The knowledge probe has two roles: (i) identify promising directions for targeted exploration within the KG, and (ii) detect potential knowledge gaps, enabling a controlled bridging mechanism that fills missing links while preserving reasoning coherence, as shown in Fig. 3.

## 4.1 DYNAMIC DISCOVERY OF KG STARTING POINTS

Exploration of a knowledge graph requires a valid starting point—an entity already present in the KG—which serves as the anchor for subsequent queries. Existing approaches typically either (i) rely on prior knowledge to directly provide a topic entity, or (ii) perform simple string-based matching against KG labels. The first strategy severely limits generalizability, while the second is prone to failure in the presence of entity aliases, abbreviations, or mismatched textual descriptions, often causing the KG exploration process to terminate prematurely.

To address these limitations, we propose a dynamically updated starting point discovery mechanism that iteratively expands a candidate set until a valid KG match is found. This mechanism maintains a set of potential starting entities and updates it in successive rounds, prioritizing direct KG entity linking before invoking GGoT's reasoning capabilities to generate new candidates with a higher likelihood of matching standardized KG identifiers.

The process begins by identifying all specific entities from the deconstructed Guidance Graph, forming an initial starting point candidate set $S = \{s_1, s_2, \ldots, s_n\}$. Each candidate $s_i$ is linked to the KG via string-based entity matching; if a corresponding KG node $n_i$ is found, it is immediately adopted as the starting point. If none of the candidates in $S$ can be linked successfully, the GGoT reasoning engine is triggered—focusing on Type II triples involving the current candidates—to perform knowledge mining. The generic entities in these triples are resolved into one or more new specific entities, which are then added to $S$ for another round of KG matching.

If the iterative expansion process yields no match, the system assumes the KG lacks relevant content and falls back to pure GGoT mode, relying on the LLM's internal knowledge to generate the answer.

## 4.2 GUIDED KG EXPLORATION VIA LLM KNOWLEDGE PROBES

Knowledge graph exploration often involves screening a vast number of potential relations and entities, where irrelevant candidates can misdirect the reasoning process. We employ *knowledge probes*—candidate entities readily obtained from LLM-based knowledge mining—to guide KG exploration and narrow the search space. This module concludes by selecting the relations and entities that meet the semantic requirements of the query via semantic consistency filtering.

First, given a Type II triple (Specific_s, Relation_r, Generic_g), we apply GGoT's knowledge mining to produce a candidate entity set $\{p\_e1, p\_e2, \ldots\}$—this set constitutes the *knowledge probe*. The specific entity Specific_s is assumed to have been successfully linked to a KG node $N$ in a previous starting point discovery round. We then retrieve all entities directly connected to $N$ via relation $r$, forming the local KG candidate set: Candidate_kg $= \{e|(N, r, e) \in \text{KG}\}$.

Next, probe verification and candidate reduction. We intersect the knowledge probe set with Candidate_KG: any probe entity found in Candidate_KG is deemed highly reliable, since it reflects consensus between the LLM and KG. To avoid omitting relevant knowledge, for each such high-

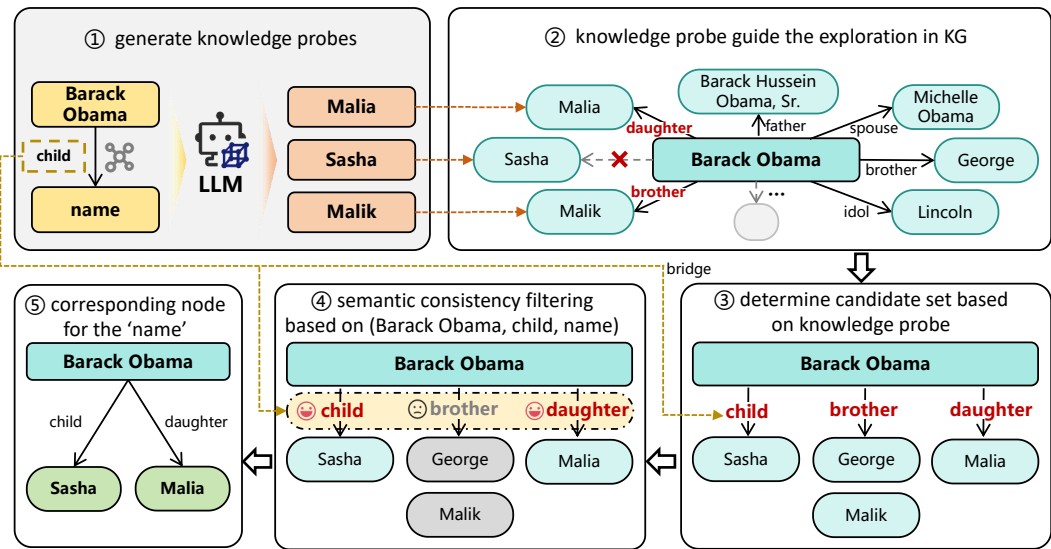

Figure 3: Example of Knowledge Probe Production and Usage in KG-LLM Collaboration. The knowledge probe Malik narrows the search scope for KG exploration, while the probe Sasha bridges the relationship between the Barack Obama node and the Sasha node in the KG. Malia is an erroneously generated probe that was filtered out during semantic consistency filtering.

confidence probe $(N, r, \text{p\_e})$, we retrieve all KG triples sharing the same $(N, r)$ context, thereby forming a *high-confidence candidate set* and significantly reducing the search space.

Finally, semantic consistency filtering. For the candidate set, semantic consistency filtering is applied to precisely identify the entity that best matches the original query semantics. The core idea is that for a qualified candidate entity $e$, its contextual relationship with node $N$ in the KG—i.e., $(N, r, e)$—should align with the semantic intent of the query triple $(\text{Specific\_s}, r, \text{Generic\_g})$.

To mitigate computational overhead caused by one-to-many relationships in the KG, use $(N, \text{relation})$ to represent the context of nodes under the same relation and complete it as $(N, \text{relation}, \text{generic})$. If the semantics of the completed $(N, \text{relation}, \text{generic})$ aligns with that of $(\text{specific}, r, \text{generic})$, then all nodes $e$ under the context of $(N, \text{relation})$ are considered semantically consistent with generic. We perform semantic consistency filtering with LLM, and the specific prompt can be found in the Appendix A5. The selected node will serve as the corresponding node for generic, thereby transforming generic into a specific entity.

## 4.3 TRUST-AWARE BRIDGING OF KG KNOWLEDGE GAPS

Even in well-curated knowledge graphs, it is common to encounter missing relational links or unrepresented facts during reasoning. To prevent reasoning breakdown in such cases, we introduce a trust-aware bridging mechanism that selectively incorporates LLM-mined knowledge while safeguarding against hallucinations. We leverage knowledge probes to detect and bridge potential knowledge gaps in the KG. If a probe entity is absent from the local candidate set Candidate_KG, two cases are possible: (i) inherent KG incompleteness—where the knowledge exists but is unrecorded in the current KG; or (ii) the probe is an LLM hallucination. To ensure trustworthiness, our bridging mechanism evaluates both *existence* and *necessity* before incorporating such knowledge.

**Existence verification and temporary bridging.** For each LLM-generated probe $p_e$ that does not match any entry in Candidate_KG, we first perform entity linking across the entire KG to verify whether $p_e$ exists as an independent node. If it exists, the knowledge is deemed highly credible, and its absence from Candidate_KG is attributed to local incompleteness (i.e., the lack of a relational edge $r$ from node $N$ to $p_e$). In this case, we dynamically construct a temporary relational edge $r$ from $N$ to $p_e$ and treat $p_e$, along with collaboratively mined results, as the node corresponding to generic_g.

If $p_e$ does not exist anywhere in the KG, it is preliminarily classified as low-credibility knowledge, potentially an LLM hallucination.

**Necessity-driven final decision.** If no eligible candidate entities remain in Candidate_KG after collaborative mining—indicating a complete absence of relevant knowledge in the KG—the low-credibility internal knowledge from the LLM becomes the sole available option. In this scenario, the necessity of adopting $p_e$ increases significantly. We then accept the LLM-provided $p_e$ as a temporary node and construct a temporary edge $r$ from $N$ to $p_e$, ensuring continuity in the reasoning chain.

As a safeguard against persistent hallucinations, any temporary nodes and edges introduced through bridging are removed if they fail to connect to credible knowledge during subsequent exploration.

### 4.4 CONSTRAINT-BASED DISAMBIGUATION IN KG-LLM COLLABORATION

Following the GGoT framework, when a generic entity appears in multiple Type II triples, cross-triple constraint-based disambiguation is performed after knowledge mining on any one of them. Consider a triple $(g \rightarrow s_1, r, s_2)$, where $g \rightarrow s_1$ indicates that $s_1$ was mined from $g$ via knowledge mining. In the original GGoT setting, the LLM is used to disambiguate $s_1$. However, in our KG-LLM collaborative framework, the LLM is reserved for disambiguating only those candidates that are absent from the KG—referred to as *low-confidence bridge nodes*.

**Credible knowledge disambiguation.** For credible knowledge present in the KG, disambiguation exploits structural connections. The triple $(g \rightarrow s_1, r, s_2)$ implies a relational link between $s_1$ and $s_2$. If $s_2$ exists in the KG, we identify correct $s_1$ candidates by retrieving all KG entities connected to $s_2$ via relation $r$, effectively filtering out irrelevant nodes. If $s_2$ is absent, we perform semantic consistency filtering over all relational edges of $s_1$ based on $(g \rightarrow s_1, r, s_2)$, selecting only edges whose relation label matches $r$. This mirrors the semantic alignment checks used in Step 3 of our framework.

**Low-confidence node handling.** For low-confidence nodes introduced via the bridging mechanism and absent from the KG, credibility can be increased if they are validated again through the LLM's internal knowledge in new contexts $(g \rightarrow s_1, r, s_2)$. Such *cross-validation* strengthens trust in the node. Conversely, nodes failing this validation are identified as hallucinations and removed using GGoT's knowledge elimination procedure.

## 5 EXPERIMENTS

### 5.1 EXPERIMENTAL SETTINGS

**Datasets and Evaluation Metrics.** To assess the effectiveness of the proposed paradigm, we employ Freebase Bollacker et al. (2008) as external knowledge base. Freebase is a large-scale, semi-structured knowledge base supported by Google, designed to organize information about millions of entities and their relations. Owing to its rich coverage, Freebase serves as a suitable external resource for our knowledge-intensive setting. We evaluate on two QA benchmarks built upon Freebase: WebQSP Yih et al. (2016) and CWQ Talmor & Berant (2018). For multi-answer questions, we report results under two evaluation metrics: (i) partial match, which considers a prediction correct if at least one gold answer is retrieved, and (ii) complete match, which requires retrieval of all gold answers.

**Baselines.** We compare our approach against eight baselines: standard prompting (IO prompt) Brown et al. (2020), chain-of-thought prompting (CoT prompt) Wei et al. (2022b), ToG Sun et al. (2024a), PoG Chen et al. (2024), StructGPT Jiang et al. (2023), KB-BINDER Li et al. (2023a), and FiSKE Tao et al. (2025b).

**Experimental Setup.** For all experiments, we fix the temperature parameter to 0 to ensure reproducibility. On the English QA datasets (WebQSP and CWQ), we adopt the original prompts from the corresponding baseline implementations.

## 5.2 PERFORMANCE COMPARISON

Table 1 presents results on WebQSP and CWQ. Our full LLM+KG collaborative method outperforms existing approaches consistently. The LLM-only GGoT framework exhibits strong sensitivity to the base model: switching from LLaMA3-8B to DeepSeek-V3 improves partial matching by 20% and more than doubles complete matching on WebQSP, with similar gains on CWQ. This highlights the importance of internal knowledge in LLMs for this setting. Notably, even with LLaMA3-8B, our collaborative framework surpasses most GPT-3.5-based methods. Performance improvements on CWQ are more pronounced than on WebQSP, likely because CWQ contains more multi-hop questions. These often expose KG incompleteness, increasing reliance on the LLM's internal knowledge, whereas WebQSP's single-hop questions depend more on the KG.

In summary, our framework achieves state-of-the-art performance across single-hop and multi-hop QA, even with smaller LLMs.

Table 1: Results for WebQSP and CWQ. In CWQ, each question has only one answer, so partial matching is equivalent to complete matching.

| Method | WebQSP | | CWQ | |
|---|---|---|---|---|
| | partial | complete | partial | complete |
| *without external knowledge* | | | | |
| IO prompt Brown et al. (2020) w/DeepSeek-V3 | 63.3 | 35.3 | 44.8 | 44.8 |
| COT Wei et al. (2022b) w/DeepSeek-V3 | 70.5 | 41.4 | 46.7 | 46.7 |
| *with external knowledge* | | | | |
| ToG Sun et al. (2024a) w/Llama3-8B | 55.6 | 32.3 | – | – |
| PoG Chen et al. (2024) w/Llama3-8B | 63.4 | 34.4 | – | – |
| FiSKE Tao et al. (2025b) w/Llama3-8B | 70.8 | 40.4 | – | – |
| StructGPT Jiang et al. (2023) w/GPT-3.5 | 72.6 | – | 54.3 | 54.3 |
| KB-BINDER Li et al. (2023a) w/GPT-3.5 | 74.4 | – | – | – |
| ToG Sun et al. (2024a) w/GPT-3.5 | 76.2 | – | 57.1 | 57.1 |
| PoG Chen et al. (2024) w/DeepSeek-V3 | 81.9 | 60.7 | 55.7 | 55.7 |
| FiSKE Tao et al. (2025b) w/DeepSeek-V3 | 82.5 | 61.1 | 50.2 | 50.2 |
| GG-explore w/Llama3-8B Tao et al. (2025a) | 79.3 | 54.1 | 56.7 | 56.7 |
| GG-explore w/DeepSeek-V3 Tao et al. (2025a) | 81.8 | 64.5 | 71.8 | 71.8 |
| GGoT (Ours) w/Llama3-8B | 52.1 | 21.2 | 35.9 | 35.9 |
| GGoT (Ours) w/DeepSeek-V3 | 72.5 | 46.4 | 53.4 | 53.4 |
| Ours w/Llama3-8B | 81.3 | 55.2 | 59.3 | 59.3 |
| Ours w/DeepSeek-V3 | **85.7** | **65.5** | **75.2** | **75.2** |

## 5.3 COMPUTATIONAL COST

Our efficiency results are summarized in Table 2. The proposed method uses a comparable number of LLM calls and tokens to FiSKE and GG-Explore, while being more efficient than PoG. On WebQSP, our approach incurs lower computational cost, as the knowledge probing mechanism mainly guides the exploration direction in its single-hop dominant setting. On CWQ, which contains more multi-hop questions, higher token consumption is required to compensate for potential KG gaps and verify connection reliability, leading to slightly higher cost than FiSKE and GG-Explore.

Table 2: Efficiency Comparison Across Datasets.

| Dataset | Method | LLM Call | Input Token | Output Token | Total Token |
|---|---|---|---|---|---|
| WebQSP | PoG | 11.3 | 6590.2 | 427.0 | 7017.2 |
| | FiSKE | 7.9 | 3079.0 | 1379.7 | 4458.7 |
| | GG-explore | 8.6 | 3264.5 | 584.0 | 3848.5 |
| | **Ours** | 8.13 | 3077.5 | 408.9 | 3486.4 |
| CWQ | PoG | 23.4 | 15483.4 | 694.8 | 16178.2 |
| | FiSKE | 9.4 | 3578.8 | 1828.7 | 5407.5 |
| | GG-explore | 10.2 | 4052.0 | 708.6 | 4760.6 |
| | **Ours** | 11.0 | 5239.5 | 619.9 | 5859.4 |

## 5.4 STUDIES ON TRUST-AWARE BRIDGING

Table 3 presents an ablation study on the Trust-aware Bridging module. The results confirm its intrinsic effectiveness: using untrusted bridging knowledge (Ours*) leads to clear performance degradation, validating the reliability of our trust judgment. The **w/o TB** denotes removing all bridging triples before answer generation, while retaining other non-bridged knowledge facts. Removing these bridging triples also significantly reduces the results, which demonstrates the effectiveness of Trust-aware Bridging.

On CWQ, 90.5% of questions were assigned trusted knowledge chains , including those with trusted bridging and those entirely derived from the KG. The remaining 9.5% of knowledge chains were discarded due to distrust. Among all questions, trusted bridging was involved in 21.8% of cases. Performance on this filtered subset (Filtered_CWQ) exceeds that of the full dataset, indicating that ~~even when bypassing untrusted chains and reverting to the base LLM,~~ answers for distrusted cases remain significantly less accurate.

Table 3: Experimental results on CWQ for trust-aware bridging (~~Filter~~ Filtered-CWQ: fully trusted QA set; Ours*: keeps untrusted bridging; w/o TB: bridging triples removed).

| Dataset | Method | Trust ratio | Bridging | LLM Call | Total Token | Partial match | Complete match |
|---|---|---|---|---|---|---|---|
| CWQ | **Ours** | 90.5% | 21.8% | 11.0 | 5859.4 | 75.2 | 75.2 |
| CWQ | **Ours*** | 90.5% | 21.8% | 11.0 | 5840.3 | 72.6 | 72.6 |
| CWQ | **Ours w/o TB** | 90.5% | 0% | 9.2 | 4114.5 | 72.2 | 72.2 |
| CWQ | **Ours* w/o TB** | 90.5% | 0% | 9.2 | 4099.1 | 69.3 | 69.3 |
| Filtered-CWQ | **Ours** | 100% | 24.1% | 11.2 | 6100.8 | 77.4 | 77.4 |

## 5.5 ABLATION STUDIES

We conducted ablation studies on three modules: (1) Dynamic Discovery of KG Starting Points — removing this module means we only determine the starting points based on the initial specific node; (2) Trust-Aware Bridging — ablating this module implies that we no longer attempt to bridge incomplete knowledge via knowledge probes; (3) Knowledge Probe — removing this module represents a reversion to existing KG-centric paradigms.

Removing each module individually leads to performance degradation, confirming their respective contributions. Notably, ablating only one module (e.g., Dynamic Discovery or Trust-Aware Bridging) results in worse performance than removing all Knowledge Probe-dependent modules together. We attribute this to the collaborative design of our framework: the absence of a single component disrupts the overall synergy, whereas a fully KG-centric baseline (removing all such modules) maintains greater internal consistency despite its limitations.

Table 4: Ablation Studies on CWQ set.

| Variant | partial match | LLM Call | Input Token | Output Token | Total Token |
|---|---|---|---|---|---|
| Ours | 75.2 | 11.0 | 5239.5 | 619.9 | 5859.4 |
| w/o dynamic discovery of KG starting points | 69.3 | 12.2 | 3850.2 | 514.7 | 4364.9 |
| w/o trust aware bridging | 69.3 | 9.2 | 3631.5 | 483.0 | 4114.5 |
| w/o knowledge probe | 70.7 | 10.7 | 4136.1 | 691.8 | 4827.9 |

## 6 CONCLUSION

Existing methods for integrating LLMs and KGs remain fundamentally KG-centric and notoriously brittle: they strictly rely on the knowledge graph as the sole source of truth to prevent hallucinations. The entire reasoning chain can collapse catastrophically even if a single link is missing in the KG. To address this issue, we propose Guidance Graph-of-Thought (GGoT), an LLM-only multi-step reasoning framework. We further extend this architecture by leveraging the internal knowledge of the LLM as a probe to guide knowledge graph exploration and bridge incomplete knowledge. Experimental results demonstrate that our method consistently outperforms existing approaches and remains effective even with smaller LLMs. Ablation studies confirm the contribution of each proposed module.

**Reproducibility Statement**  Our main code has been submitted with full algorithm logic and annotations. The utils.py file referenced handles LLM calls and database operations, which do not affect the algorithm. LLM prompts are included in the appendix, and database function returns are noted in the code. With the provided code and appendix prompts, our method can be readily reproduced.

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

# A APPENDIX

## A.1 LLM USAGE STATEMENT

The large language model was only used for grammar checking and polishing, and nothing else.

## A.2 PROMPT TEMPLATES

In this subsection, we introduce all prompt templates used in this paper. The complete prompt A1 and structure prompt A2 are employed for constructing the guidance graph. The internal prompt A3 is used in GGoT to mine the internal knowledge of the LLM, while the specific_internal prompt A4 serves for disambiguation. The prune prompt is A5 applied for semantic consistency filtering, and answer_by_triples A6 is designed for generating answers based on knowledge chains.

Table A1: complete_prompt.

| complete | Rephrase the question as a statement, ensuring:
1. All explicit and logically implied information is included (e.g., location, time, scope if hinted in the question).
2. Split compound noun phrases into individual entities.
3. Classify each entity as generic/specific.
Do not answer the question.
question: What team did Payton Manning's father play for?
answer:
statement: Identify the team Payton Manning's father played for.
entities:
  - team (generic)
  - Payton Manning (specific)
  - father (generic)
question: what did James K. Polk do before he was president?
answer:
statement: Identify the job of James K. Polk before he became president.
entities:
  - job (generic)
  - James K. Polk (specific)
  - president (generic)
question: Where did the "Country Nation World Tour" concert artist go to college?
answer:
statement: Identify the college attended by the concert artist of the "Country Nation World Tour."
entities:
  - college (generic)
  - Country Nation World Tour (specific)
  - concert artist (generic)
Now answer with the format of the example above. Be brief and precise.
question: {}
answer: |
|---|---|

Table A2: structure_prompt

| structure | Please analyze the relationships between the following entities in the given sentence.
Represent each relationship as a triple in the format (subject, relation, object).
Important Rules:
- Do not alter the provided keywords. Use them exactly as given (no paraphrase, no pluralization).
You may add other words only if they appear in the sentence.
- If a generic keyword and another keyword refer to the same entity,
the generic keyword must be treated as a relation rather than an entity.
- Ensure the triples are logically connected.
sentence: Identify the father of Keyshia Cole.
generic keywords: father
specific keywords: Keyshia Cole
answer:
triples: [(father, of, Keyshia Cole)]
sentence: Identify the main trading partner of China that appointed Abdelaziz Bouteflika to a governmental position.
generic keywords: main trading partner; governmental position
specific keywords: China; Abdelaziz Bouteflika
answer:
triples: [(main trading partner, of, China), (main trading partner, appointed, Abdelaziz Bouteflika),
(Abdelaziz Bouteflika, appointed to, governmental position)]
Now answer with the format of the example above. Be brief and precise.
sentence: {}
generic keywords: {}
specific keywords: {}
answer: |
|---|---|

Table A3: internal_prompt.

| | |
|---|---|
| internal | Task: Infer the entity that fills the target slot in the triples using only the semantics of the relation and general world knowledge.

Critical rules:
- non_target_context contains information about the other side of the triple (the complementary entity), NOT the target.
- Never output any string that appears as a key or value in non_target_context.
If you include any item from non_target_context in your answer, it is wrong.
- Output concrete, specific entities that can occupy the target slot
(e.g., for target=award: award names; for target=year: a 4-digit year).
- Multiple correct answers may exist; list the most likely 1–5.
- If you cannot reasonably infer any specific target, output []
- Output format must be exactly: [answer1, answer2, ...] with no other text.

Examples:
triples: [(Marie Curie, winner of, award)]
target slot: award
non_target_context: {}
answer: [Nobel Prize]

triples: [('team', 'lastWonWorldSeries', 'year')]
target slot: year
non_target_context: {{'team': 'San Francisco Giants'}}
answer: [2014]

triples: [('director', 'directed', 'film')]
target slot: film
non_target_context: {{'director': 'Greta Gerwig'}}
answer: [Lady Bird, Little Women]

triples: [('author', 'wrote', 'book')]
target slot: book
non_target_context: {{'author': 'Haruki Murakami'}}
answer: [Norwegian Wood, Kafka on the Shore]

Now answer this:
triples: {triples}
target slot: {target_entity}
non_target_context: {previous_knowledge}
answer: |

Table A4: specific_internal_prompt.

| | |
|---|---|
| specific_internal | Please rely on your own knowledge to find the answer to the target entity based on the given triples and previous knowledge.
There may be more than one answer. Here are some examples.

triples: [(Marie Curie, winner of, award)]
previous knowledge: {{award: [Nobel Prize, Fields Medal, Turing Award]}}
answer: [Nobel Prize]

triples: [(programming language, named after, comedy group)]
previous knowledge: {{programming language: [Python, C++]}}
answer: [Python]

Now answer with the format of the example above.
triples: {triples}
previous knowledge: {previous_knowledge}
answer: |

Table A5: prune_prompt.

| | |
|---|---|
| prune | Select the most semantically matching relation from the candidate relations to replace the given relation in the theme information while maintaining its essential meaning. Rate it on a scale of 0 to 10. Be brief and precise.

Below is an example:
theme information: main spoken language of country
relation: of
candidate relations: {
   'language.human_language.main_country',
   'language.human_language.human_language',
   'language.human_language.language_family',
   'language.human_language.iso_639_3_code',
   'base.rosetta.languoid.parent',
   'language.human_language.writing_system',
   'base.rosetta.languoid.languoid_class'
}
assess:
- 'language.human_language.human_language' can perfectly replace the relation, (9) score.
- 'language.human_language.main_country' can replace the relation, (8) score.
- other relations are not related to any information in the sentence, (0) score.

Now answer with the format of the example above. Be brief and precise.
theme information: {}
relation: {}
candidate relations: {}
assess: |

Table A6: answer_by_triples_prompt.

| | |
|---|---|
| answer_by_triples | Please answer the question. Triples are available for reference.
If there is not enough information in the triples, please answer with your own knowledge.

question: Which place is the madam satan located?
triples: [('madam satan', 'film.film.country', 'the USA'),
('madam satan', 'film.film.language', 'English')]
answer: 'madam satan' is located in 'the USA'.

question: {}
triples: {}
answer: |

## A.3 GUIDANCE GRAPH CONSTRUCTION RULES

1. Specific keywords exclusively serve as graph nodes (i.e., entities in triples) rather than edges.

2. When a generic keyword co-refers to the same entity as a specific keyword, it functions as a triple's relation rather than an entity.

3. For two distinct generic keywords referencing the same entity, one must be assigned as the relation in the triple.

4. For associated keywords referring to different entities, we construct triples where the keywords become head/tail entities and their association forms the relational edge.

