# OpenReview forum: "Bridging Gaps with Dynamic Knowledge Probes: Robust LLM–KG Collaborative Reasoning"
_ICLR.cc/2026/Conference — ICLR 2026 Conference Desk Rejected Submission_

### Official Review · Reviewer_i9nf · 2025-10-18

**Soundness:** 3
**Presentation:** 2
**Contribution:** 3
**Rating:** 6
**Confidence:** 3

**Summary:**

Large Language Models (LLMs) excel at language tasks but face limits from static knowledge, hallucinations, and opaque reasoning. Existing agent methods over-rely on Knowledge Graphs (KGs) and ignore LLM priors, failing when links are missing. They propose a KG–LLM collaborative framework that treats the LLM’s knowledge as dynamic probes—produced by a Guidance Graph of Thought (GGoT) from partial triples—to steer KG search, detect gaps, and perform trust-aware bridging with existence/necessity checks, followed by cross-triple constraint disambiguation. Experiments on multiple benchmarks show consistent gains over prior methods, with ablations confirming each component’s contribution.

**Strengths:**

1. Dynamic KG–LLM Collaboration – They combine Knowledge Graph verification with LLM reasoning, maintaining robustness even under KG incompleteness.

2. Guidance Graph of Thought (GGoT) – They introduce a modular reasoning backbone that enables transparent, multi-step reasoning and seamless KG integration.

3. Trust-Aware Knowledge Bridging – They ensure reliability by verifying the existence and necessity of LLM-derived entities before integration, preventing hallucination propagation.

**Weaknesses:**

1. The paper presentation is lacking. First of all, there are too many line breaks. In the abstract, line breaks are generally not used, and in particular, Section 5.2 feels cluttered with unnecessary line breaks.

2. The motivation and methodology of this study do not seem to be fully aligned. Early in the paper, it is stated that the use of a KG alongside an LLM aims to mitigate issues such as static knowledge, potential hallucinations, and opaque reasoning processes in LLMs. However, the method describes filling the incomplete parts of the KG with LLM-generated information, which does not sound logically consistent. It would be better to revise the motivation so that it resonates more clearly with readers.

3. The datasets used in the experiments seem to be favorable to this framework. Rather than relying only on WebQSP and CWQ, testing in settings where LLM hallucinations are more likely to occur would be more effective in demonstrating the generalizability of the proposed framework.

**Questions:**

The points I wrote under Weaknesses.

---

> ### Author Response · Authors · 2025-11-20
> **Response to Reviewer i9nf (1/1)**
>
> ## Weakness 1
> We thank the reviewer for pointing out the presentation issues, especially the excessive use of line breaks in the abstract and Section 5.2. We acknowledge that this formatting may have disrupted the reading flow. In the revised manuscript, we have removed unnecessary line breaks, ensured that the abstract is presented as a continuous paragraph, and streamlined Section 5.2 for better readability and coherence.
>
> ## Weakness 2
> We appreciate the reviewer’s comment regarding the possible misalignment between our stated motivation and the described methodology. We acknowledge that the wording in our original introduction may have led to the impression that LLM-generated content is directly injected into the KG as authoritative facts.
>
> In our framework, the LLM is not used as an unconditional source of truth. It produces probes—candidate entities—treated explicitly as hypotheses to guide exploration and identify KG incompleteness. These hypotheses are cross-checked using the procedures described in Sections 4.3 and 4.4, where they are verified against existing KG structure before being used or incorporated. This ensures our motivation (leveraging the KG’s verifiability while addressing its incompleteness) and methodology are fully consistent.
>
> To clarify this point, we have revised the introduction to explicitly define the LLM’s role as a hypothesis generator and to describe the trust-aware bridging process. In addition, our second case study, presented in the public OpenReview reply , illustrates the multi-round handling of hallucinated probes. This concrete example demonstrates that unverifiable LLM content is never unconditionally injected into the KG.
>
> ## Weakness 3
> We thank the reviewer for the comment on dataset choice.
> WebQSP and CWQ are standard and widely adopted benchmarks in knowledge graph question answering research, chosen to enable direct comparison with prior work. These datasets are not “favorable”: they present diverse query structures, including multi-hop reasoning and rare entities, which can trigger hallucinations in LLMs.
>
> As shown in Section 5.4, across all evaluation questions, 9.5% result in bridging candidates whose trustworthiness cannot be determined after exploration and must therefore be answered directly by the LLM without KG verification. In the revised manuscript, we additionally report that 21.8% of all questions involve bridges that are successfully verified and retained as trustworthy. The remainder are either answerable without bridging, or involve bridges rejected as untrustworthy (i.e., hallucinations), for which we did not record the exact proportion. Consistent with our second case study, these figures confirm that hallucinations arise frequently in these benchmarks, and our framework systematically addresses them via trust-aware verification.
>
> We agree that evaluating on datasets with higher hallucination rates—such as open-domain web QA or domains with more ephemeral knowledge—would further strengthen empirical claims. As future work, we aim to extend evaluations to such settings to demonstrate broader generalizability.

---

> > ### Author Response · Authors · 2025-12-03
> > **Response to Reviewer i9nf (Update)**
> >
> > Thank you for the insightful suggestion regarding evaluation on settings where hallucinations are more likely to occur. Following your comment, we conducted additional experiments on Wikidata, which presents substantially harsher conditions than the original Freebase-based benchmarks.
> >
> > Unlike ToG—whose reported Wikidata results rely on a clean and curated dump—our rebuttal-time experiments necessarily use the official Wikidata API, which intermittently returns incomplete results. This setup introduces additional incompleteness and therefore amplifies hallucination-triggering scenarios for our method. Despite these unfavorable conditions, our approach remains robust and continues to outperform prior methods under comparable cross-KG settings.
> >
> > To avoid repeating the shared rebuttal, we do not include experimental tables here, but all results are summarized in the general response. Collectively, they demonstrate that our framework maintains strong performance even in environments with significantly higher hallucination risk, directly validating the reviewer’s concern and reinforcing the generalizability of our approach.

---

### Official Review · Reviewer_v2w7 · 2025-10-31

**Soundness:** 2
**Presentation:** 1
**Contribution:** 1
**Rating:** 2
**Confidence:** 5

**Summary:**

This paper introduces Dynamic Knowledge Probes (DKP), a novel framework for robust collaboration between Large Language Models (LLMs) and Knowledge Graphs (KGs) under conditions of graph incompleteness. Unlike conventional KG-authoritative methods, which strictly rely on the KG as the sole source of truth, DKP repositions the LLM as an active knowledge probe generator. DKP bridges the gap between question complexity and knowledge incompleteness by: (1) Guided KG Exploration via LLM Probes: Using lightweight, LLM-generated entities to steer KG traversal toward semantically relevant subgraphs. (2) Trust-Aware Bridging: Performing existence and necessity checks to safely integrate LLM-inferred entities when KGs are incomplete. (3) Constraint-Based Disambiguation: Enforcing cross-triple consistency via KG structure for credible nodes and LLM validation for low-confidence ones. Experiments on WebQSP and CWQ show that DKP achieves state-of-the-art accuracy.

**Strengths:**

The paper proposes a clear paradigm shift: instead of treating the LLM as a passive query generator, it becomes an active collaborator capable of proposing missing links under verifiable control. This insight directly addresses a long-standing bottleneck of KG incompleteness. The proposed Guidance Graph of Thought (GGoT) is a coherent, interpretable framework. It cleanly separates reasoning stages (triplet resolution, probing, bridging), allowing transparent integration of KG and LLM modules.

**Weaknesses:**

1. Missing highly relevant literature. The paper identifies one key bottleneck in LLM reasoning, that the internal static knowledge and external knowledge base can be both sparse, which is discussed in [1]. The proposed retrieval method that focuses on semantically similar subgraphs and use LLM internal knowledge to bridge the knowledge gap also overlaps with [1].

2. Too much white space in the body of the paper.

3. See Questions for other technical weakness.

[1] GIVE: Structured Reasoning of Large Language Models with Knowledge Graph Inspired Veracity Extrapolation

**Questions:**

1. Could the authors provide a detailed comparison with [1]? In terms of methodology and empirical results?

2. How sensitive the proposed method is to different sparsities of KG? It is necessary to evaluate the performance of the proposed method on KGs with different sparcities (edge ratio), to prove its effectiveness in "BRIDGING GAPS" between question complexity and knowledge incompleteness.

3. Does the proposed method generalize to different domains? As the proposed method relies highly on LLM-generated probe, its beneficial to test its generalizability to corner domains (scientific for example) where the  model's internal knowledge is even more limited, especially considering the fact that "switching from LLaMA3-8B to DeepSeek-V3 improves partial matching by 20%
and more than doubles complete matching on WebQSP".

4. The experiment setup is not consistent with the main claim about bridging gaps between incomplete KG and complex QA. It does not make sense to compare partial or complete gold answer retrieval. Instead, comparison should be focused on how the competing baselines deal with incomplete KG and conduct correct reasoning to answer the question that the gold context is not accessible.


[1] GIVE: Structured Reasoning of Large Language Models with Knowledge Graph Inspired Veracity Extrapolation

---

> ### Author Response · Authors · 2025-11-20
> **Response to Reviewer v2w7 (1/2)**
>
> **Dear Reviewer,**
>
> Thank you for your insightful and constructive comments. Your question on the relationship between our work and GIVE [1] touches the core motivation of our study, and we believe part of the misunderstanding may stem from differing assumptions. Before responding to your specific points, we would like to **clarify our motivation**.
>
> We argue that **missing knowledge in KGs is inevitable**, even in very dense graphs, and not solely due to sparsity. Existing KG-authoritative methods are **brittle** in such cases: a single missing link can cause the reasoning chain to fail. **Our method** is designed to maintain reasoning continuity despite missing links. *As shown in Figure 1*, when **The Beatles** node is absent, traditional methods halt after *John Lennon*; our approach uses the **LLM’s internal knowledge** to generate probes, add the missing node, and proceed in the KG to entities such as *Abbey Road*.
>
> **GIVE**, by contrast, targets **sparse KB** scenarios (“external knowledge base can be sparse”), focusing on discovering associations in sparse graphs. We focus on **dense KG** scenarios, aiming for **robustness** while meeting strict **accuracy** and **efficiency** requirements. Applying GIVE to dense graphs (e.g., **Freebase** with 1.9 billion triples) would incur prohibitive computational costs. Further details will be provided in our response to your first question.
>
> ## Weakness 1 & Question 1
>
> ### 1. Overall Workflow
> GIVE adopts a *many-to-few* strategy: it first retrieves **all possible entities** from the KG and then filters them down.
> Our approach follows a *few-to-many* exploration process: starting from **a single KG starting point** and iteratively expanding outward.
>
> - **GIVE**:
>   1. Extracts entities and relations from the question.
>   2. Retrieves candidate nodes from the KG via **vector similarity**, grouping them into subsets.
>   3. Within each group, uses an LLM to generate potential relations.
>   4. Forms an inter-group relation candidate set from (extracted relations + KG relations + LLM-generated descriptions) and applies LLM-based filtering.
>   5. Iteratively produces the final answer from the filtered knowledge.
>
> - **Ours**:
>   1. Extracts entities and relations from the question to build a **Guidance Graph**.
>   2. Locates a starting point in the KG via a dynamic entry selection mechanism.
>   3. Iteratively explores the KG outward to resolve the Guidance Graph’s pseudo-triples, using **knowledge probes** to bridge incompleteness.
>   4. Constructs a reasoning chain from the explored knowledge and answers the question.
>
> ---
>
> ### 2. Key Differences in Design
> 1. **Integration of extracted information**
>    - GIVE uses extracted entities for similarity search and extracted relations for LLM relation judgments, but these are *used independently*.
>    - Our method builds a unified Guidance Graph from extracted info, guiding every exploration step and probe generation.
>
> 2. **Purpose of relation supplementation**
>    - GIVE supplements relations to identify connections *among already-retrieved candidates*.
>    - We supplement relations to **continue KG exploration** and acquire *new* knowledge.
>
> 3. **Handling missing entities**
>    - GIVE assumes candidate entities are present in the KG before relation supplementation.
>    - Our approach can **bridge missing entities** via LLM probes and continue exploration.
>
> 4. **LLM dependency and hallucination control**
>    - GIVE relies entirely on LLM for inner-group relation generation and inter-group relation veracity judgment, with no measures to prevent hallucinations.
>    - Our Sections 4.3 and 4.4 introduce **multiple safeguards** to mitigate LLM hallucination impact.
>
> ---
>
> ### 3. Experimental Context
> 1. **Dataset scale and challenges**
>    - GIVE operates on small KGs (≤ 2 M triples), with low efficiency demands and minimal noise.
>    - Our experiments use **Freebase** (≈ 1.9 B triples), requiring high efficiency and robustness against heavy noise.
>      *Example:* In Case 1 of our case study, the first round selects the correct link from **498 different relations** of *France*; the third round selects the correct entity from **15,184 candidates**.
>
> 2. **Computational cost**
>    - From GIVE’s Table 8, its computational overhead is much higher than ToG.
>    - Our Table 2 reports computational cost comparisons with ToG, showing **significantly lower overhead**.
>
> ---
> We have added a brief discussion of GIVE in the revised manuscript

---

> ### Author Response · Authors · 2025-11-20
> **Response to Reviewer v2w7 (2/2)**
>
> ## Question 2
> We appreciate your suggestion regarding sensitivity analysis under varying sparsities of the knowledge graph (KG). We agree that such analysis can be valuable for assessing robustness. Here we would like to clarify our motivation and explain how our current experimental setting already addresses your concern.
>
> Our work focuses on mitigating the brittleness of existing KG-authoritative methods under real-world knowledge incompleteness, rather than explicitly maintaining performance across artificially controlled sparsity levels. In practice, a more meaningful sparsity-sensitivity evaluation is to consider a realistic large-scale KG (e.g., Freebase), whose internal connectivity distribution is inherently uneven: some entities are densely connected, while others are extremely sparse.
> Under this setting, different questions naturally involve subgraphs with widely varying effective sparsity. Our state-of-the-art results on WebQSP and CWQ—both based on Freebase—demonstrate that the proposed method effectively handles the challenges posed by subgraphs of different sparsities in a realistic environment. We believe such evidence is more representative and convincing than evaluations on artificially pruned, small-scale KGs with uniform sparsity settings.
> To address your concern more directly, we have also added two case studies in our public OpenReview reply:
> - **case 1**: involves a highly dense subgraph with hundreds of relations and tens of thousands of entities.
> - **case 2**: involves a very sparse subgraph with only a few dozen relations and entities.
> These analyses further illustrate the robustness of the proposed method across a wide range of natural sparsity levels in real-world KGs.
>
> ## Weakness 2
> Thank you for pointing this out. We have revised the manuscript to reduce the excessive white space in the body of the paper.
> ## Question 3
> We would like to clarify that the proposed **probe mechanism** in our method serves as an *enhancement* rather than a *dependency*. Even without any probes, our approach can still exploit the LLM’s semantic reasoning capability together with the structural information of the KG to explore correct knowledge paths. This is demonstrated in the **ablation experiments** in Section 5.5, where the removal of probes still preserves competitive performance.
>
> Regarding the reviewer’s mention of
>  “switching from LLaMA3‑8B to DeepSeek‑V3 improves partial matching by 20% and more than doubles complete matching on WebQSP”.
> The full statement in our paper refers to the **LLM‑only GGoT baseline**, which relies solely on LLM internal knowledge and is thus highly sensitive to the choice of base model. For our extended GGoT framework **with knowledge probes**, the performance change when switching from LLaMA3‑8B to DeepSeek‑V3 is comparatively moderate:
> - **WebQSP**: complete matching improves from *55.2* to *65.5*
> - **CWQ**: complete matching improves from *59.3* to *75.2*
>
> These improvements partially benefit from the richer internal knowledge of DeepSeek‑V3, but importantly also from its **enhanced semantic capabilities**.
>
> Furthermore, in the newly added **case study (Case 2)**, the LLM‑generated probes from internal knowledge completely fail in the first round, and only one probe is valid in the second round. Even under this probe‑scarce scenario, our approach successfully finds the correct answer by leveraging **semantic matching** and **KG structure exploration**, indicating strong resilience to imperfect probes and supporting effectiveness in domains where LLM internal knowledge is limited.
>
> **Conclusion:**
> Overall, the results suggest that our method *does* generalize beyond its probe component, and can perform well in less-covered domains — including **scientific ones** — by relying on **semantic reasoning** and **KG structure** rather than requiring high‑coverage internal LLM knowledge.
>
> ## Question 4
> We appreciate your suggestion that the comparison should focus on how competing baselines deal with incomplete KGs and perform correct reasoning when the gold context is not accessible.  As mentioned in our earlier response, we have already provided a detailed explanation regarding the consistency between our main claim and the experimental setup.
>
> Regarding your point on focusing the comparison, existing KG-authoritative methods do not address incomplete KGs, and the GIVE framework you refer to differs substantially from our experimental scenario. These differences make a direct empirical comparison infeasible. However, we have reported in Section 5.4 some data and analyses related to the “bridging” process, which illustrate how our method fills missing links and continues reasoning without access to the gold context.

---

### Official Review · Reviewer_e5Aj · 2025-11-03

**Soundness:** 4
**Presentation:** 4
**Contribution:** 4
**Rating:** 8
**Confidence:** 3

**Summary:**

The paper presents **Dynamic Knowledge Probes over a Guidance Graph-of-Thought (GGoT)** for **LLM–KG collaborative reasoning**. Starting from a structured “guidance graph” built from the question, the method iteratively converts **generic** entities to **specific** ones (Type II→Type I triples) using the LLM’s parametric knowledge, while interleaving **KG verification**. The key contributions are: (i) **knowledge probes**—LLM-suggested candidate entities that guide local KG exploration; (ii) **trust-aware bridging**—existence and necessity checks that temporarily insert LLM-derived entities/edges when the KG appears incomplete, with later pruning if they fail to connect; and (iii) **constraint-based disambiguation** across triples to enforce consistency (preferring KG structural checks and reserving LLM only for low-confidence “bridge” nodes). Experiments on **WebQSP** and **CWQ** (Freebase) show improvements over ToG/PoG/FiSKE/StructGPT and GG-explore in both partial and complete match; efficiency is comparable to or better than prior agents on WebQSP and slightly higher on CWQ; ablations highlight the benefit of trust-aware bridging and probes.

**Strengths:**

* **Well-motivated robustness target.** Clearly identifies brittleness of KG-authoritative agents under incompleteness and proposes a principled collaboration scheme.
* **Clean, extensible backbone.** GGoT’s fact-level interfaces decouple steps and make KG insertion points natural.
* **Trust-aware bridging is explicit.** Existence/necessity checks and later pruning provide a defensible safety valve against hallucination.
* **Empirical gains + ablations.** SOTA-level results on WebQSP/CWQ (esp. with DeepSeek-V3); efficiency roughly on par with strong agents; ablations on bridging/probes and “trusted subset” analysis.

**Weaknesses:**

1. **Risk of evaluation conflation.** Temporary **edge insertion** effectively augments the KG with LLM knowledge; without strict auditing, improvements may partly reflect parametric recall rather than better KG reasoning. Authors should report **rates, locations, and impact** of bridging (how often temporary edges persist vs. get pruned; effect size per category).
2. **LLM-dependent validators.** Semantic consistency filtering and low-confidence node validation rely on prompts (A5), which can be brittle and model-biased. A **KG-executability** or **type-constraint** signal would be a stronger arbiter than LLM scoring alone.
3. **Limited statistics.** Main tables omit **CIs/multi-seed**; given stochastic components, uncertainty should accompany all metrics and efficiency numbers.
4. **Scope and generality.** Only Freebase (WebQSP/CWQ). No results on **Wikidata/DBpedia** or **text+KG** regimes, where aliasing and schema variance are larger.
5. **Entity-linking details.** Starting-point discovery iterates EL and GGoT, but matching rules/ambiguity handling are under-specified; errors here could cascade.
6. **Ablation interpretation.** Table 4 shows non-monotone cost/accuracy tradeoffs when removing single modules; more granular diagnostics (e.g., which question types benefit) would clarify interactions.

**Questions:**

1. **Bridging audit.** Please report: fraction of questions where bridging is triggered; proportion of **temporary nodes/edges** that survive pruning; and accuracy deltas **with vs. without** those edges on the same items.
2. **Validator alternatives.** Can you replace/augment LLM-based semantic filtering with **schema/type constraints**, **range/domain checks**, or **executed path validation**? Any accuracy drop if you remove LLM scoring in A5?
3. **Generality.** Any results or qualitative studies on **Wikidata** or **Freebase+Wikipedia** (text+KG) to assess robustness to aliasing and missing edges?

---

> ### Author Response · Authors · 2025-11-20
> **Response to Reviewer e5Aj  (1/2)**
>
> ## Weakness 1 & Question 1
> Thank you for raising this important concern regarding the risk of conflating LLM parametric recall with genuine KG reasoning.
> In Section 5.4, we report that 90.5% of CWQ questions were answered using trusted knowledge chains—either fully derived from the original KG or augmented with verified bridging edges—while the remaining 9.5% involving untrusted bridges were discarded. Crucially, excluding untrusted chains improves accuracy from 72.6% (without filtering) to 75.2% (with trust-based pruning), confirming our trust mechanism filters out harmful LLM hallucinations.
>
> Moreover, to assess the positive contribution of trusted bridges, we conducted an ablation: when we remove only the trusted bridging edges (i.e., restrict reasoning to the original KG), performance drops from 75.2% to 72.2%, demonstrating that these verified temporary edges genuinely enhance reasoning rather than merely injecting parametric knowledge.
>
> Following your suggestion, we now additionally report that 21.8% of all questions involve at least one trusted bridging edge. This statistic—added to Section 5.4—quantifies how frequently temporary edges are retained after auditing and underscores that our gains stem from validated KG augmentation, not unfiltered LLM recall.
>
> ## Weakness 2 & Question 2
> We appreciate the reviewer’s thoughtful suggestion. Our design already mitigates over-reliance on LLM-based semantic filtering in several ways.
>
> First, the semantic consistency step (A5) is a selection task—comparing candidate relations against Type-II triples—rather than open-ended generation, which limits hallucination and model bias.
> Second, we extensively use KG structural information for disambiguation (Section 4.4). While schema/type or domain/range constraints are useful, they often require strong priors or additional LLM calls, which we aim to avoid to maintain broad applicability.
>
> Low-confidence nodes—identified in probe verification (Section 4.2)—are typically discarded unless they pass two stringent checks (Section 4.3). Even then, they may be pruned later if they fail to connect to credible nodes (Section 4.4).
>
> We’ve added two case studies in our public OpenReview reply: one highlights structural disambiguation, and the other shows how low-confidence probes are handled conservatively.
> Thus, LLM scoring in A5 operates within a robust validation pipeline that prioritizes KG structure and cautious verification, minimizing both LLM dependency and error propagation.
>
> ## Weakness 4 & Question 3
> Our current setting differs from text+KG in an important aspect: existing text+KG methods (e.g., Microsoft GraphRAG) typically construct the KG directly from text, so entities and relations are tightly intertwined with textual passages. In contrast, the KG in our experiments is built independently and does not contain text-level contextual information. This makes our scenario closer to the “independent multi-source knowledge” setting, where KG and text are not automatically aligned. The proposed GGoT framework is designed to be extensible to heterogeneous knowledge sources, and our current experiments demonstrate its ability to effectively integrate independent KG knowledge with LLM internal knowledge; incorporating textual corpora is part of our ongoing work. Regarding Wikidata, we have started preliminary deployment and experimentation. This process will take some time, but we will make every effort to include results in the revised or camera-ready version.
>
> ## Weakness 3
> We agree that reporting uncertainty estimates is important, especially for settings with stochastic components. In our case, repeated full-dataset evaluations are impractical because the core reasoning steps rely on LLM APIs (e.g., DeepSeek-V3) whose internal sampling randomness cannot be fully disabled and each full run incurs substantial API costs.
> To mitigate variance under these constraints, we used fixed prompts and deterministic decoding parameters wherever possible (temperature = 0, top-p = 1). And we conducted subsample stability checks by re-running the complete pipeline three times on a random 100-question subset of CWQ. The results are:
> | Model       | Run 1 | Run 2 | Run 3 | Mean ± Std       |
> |-------------|:-----:|:-----:|:-----:|:----------------:|
> | DeepSeek-V3 | 74.1  | 76.7  | 76.3  | **75.70 ± 1.4** |
> | LLama3-8B   | 55.2  | 59.4  | 56.8  | **57.13 ± 2.12** |
>
> We observe that DeepSeek-V3 yields more stable results across runs, while LLama3-8B shows higher variability. We hypothesize this is because, in our method, model-internal knowledge and semantic reasoning capabilities complement each other. DeepSeek is comparatively strong in both aspects, so when one factor is slightly perturbed, the other can compensate.

---

> ### Author Response · Authors · 2025-11-20
> **Response to Reviewer e5Aj (2/2)**
>
> ## Weakness 5
> In our implementation, entity linking in the starting-point discovery stage is performed through straightforward character-string matching on named entities (i.e., via SPARQL SELECT queries against the knowledge base). Given that any successfully linked starting point can initiate the process, the retrieval of subsequent nodes is confined primarily to entities linked to established ones.
> Unlike many existing approaches that rely on fixed, manually specified starting points as prior knowledge, our dynamic starting-point strategy requires no such assumptions and can adapt to unseen input. Empirically, it achieves comparable success rates while being more practical in scenarios where prior information is unavailable. We will add these clarifications to the manuscript.
>
> ## Weakness 6
> Thank you for this insightful suggestion. We agree that a more granular diagnostic (e.g., by question type) would greatly help clarify the module interactions noted in Table 4. Due to the time constraints of the rebuttal, conducting this extensive analysis at this stage is challenging. Therefore, we will perform a comprehensive analysis based on question types and include it in the final version of the paper.
> Regarding the non-monotonic trade-offs, our initial analysis suggested that removing a single module disrupts the overall synergy, leading to performance degradation, while removing all modules results in a more internally consistent, albeit limited, KG-centric baseline. We will further investigate and validate this hypothesis within the supplementary analysis across different question types.

---

> > ### Author Response · Authors · 2025-12-02
> > **Response to Reviewer e5Aj (Update)**
> >
> > Thank you again for raising this important concern regarding the scope and generality of our evaluation. As mentioned in our earlier response, we had begun running experiments on Wikidata. We are glad to report that we have completed an initial round of experiments before the rebuttal deadline. The detailed results and analysis are now included in the *Additional Experiments on Wikidata* section of the general response.
> >
> > These new results on Wikidata — a significantly larger and noisier KG with substantial aliasing and schema variation — show that our method maintains strong performance and generalizes well beyond Freebase-based benchmarks. This directly addresses the concern about the method’s applicability to more challenging, real-world KG settings.

---

### Official Review · Reviewer_9oaZ · 2025-11-04

**Soundness:** 3
**Presentation:** 3
**Contribution:** 3
**Rating:** 4
**Confidence:** 5

**Summary:**

This paper introduces Dynamic Guidance Graph of Thought (D-GGoT), a framework that enables collaboration between LLMs and knowledge graphs (KGs) for multi-hop reasoning under graph incompleteness. Instead of letting the LLM generate full answers or relying solely on KG structure, D-GGoT decomposes the query into triple-level reasoning steps, allowing the LLM to suggest candidate entities/relations and verify them through trust-aware bridging and constraint-based disambiguation. The system dynamically discovers starting points when entity linking fails and iteratively refines generic triples into more specific ones. Experiments on WebQSP and ComplexWebQuestions (CWQ) show that D-GGoT achieves superior accuracy, particularly on multi-hop queries, while maintaining reasonable efficiency.

**Strengths:**

The paper clearly diagnoses the brittleness of KG-authoritative agents: one missing KG link can collapse the whole reasoning chain. This motivates letting LLMs “assist” rather than being fully suppressed. The Guidance Graph of Thought (GGoT) turns queries into triples with three specificity types and iteratively converts generic→specific, exposing a fact-level interface that interleaves KG verification without entangling prior textual rationales. This is a neat, extensible backbone. Three concrete collaboration mechanisms. (1) Guided KG exploration via LLM knowledge probes; (2) trust-aware bridging with existence/necessity checks; (3) cross-triple, constraint-based disambiguation that prefers KG structure for credible nodes. Together they operationalize “LLM assists KG” instead of replacing it.

**Weaknesses:**

While the paper presents an interesting and well-motivated framework for integrating LLMs with knowledge graphs, several concerns remain regarding evaluation scope and methodological transparency:

1. The experiments are primarily conducted on Freebase-based datasets (WebQSP, CWQ). To better demonstrate generality, the authors should include results or analyses on more widely used KGs such as Wikidata or DBpedia ideally with noisy or imperfect entity linking to assess robustness.

2. The comparison table involves multiple base LLMs (e.g., GPT-3.5, DeepSeek, LLaMA3-8B). It would be clearer to group or standardize results by model family or size for consistent comparison.

3. Several internal procedures—such as semantic consistency filtering, triple pruning, and relation validation—depend heavily on LLM responses. This reliance may introduce hallucination or alignment drift, potentially leading to incorrect triple acceptance. An empirical analysis or mitigation strategy for such hallucination errors would strengthen the paper.

4. The paper does not report time, token, or computational cost compared to other baselines. Given that the proposed method includes multiple iterative calls and reasoning stages, an efficiency comparison is essential for assessing practical utility.

**Questions:**

See weaknesses above.

---

> ### Author Response · Authors · 2025-11-20
> **Response to Reviewer 9oaZ  (1/2)**
>
> ## Weakness 1
>
> We thank the reviewer for the suggestion to evaluate our framework on other widely used KGs such as Wikidata or DBpedia, which could better demonstrate its generalizability and robustness under noisy or imperfect entity linking.
> Our current experiments focus on Freebase-based datasets (WebQSP, CWQ) because all competitive baselines in this domain are reported on Freebase, and its static nature ensures fair, direct comparison. In contrast, Wikidata and DBpedia are continuously evolving, making direct reproduction and comparison with prior work potentially unfair.
> Freebase is nevertheless a large-scale KG (≈1.9B triples) and exhibits substantial noise. For example, in our first case study (shown in our public OpenReview reply), during the first reasoning round, the entity France had 498 edges of different relation types, and our probing module efficiently identified the single correct edge. In the third reasoning round, our credible knowledge disambiguation component selected the unique correct entity among 15,184 candidates by leveraging KG structural constraints. These examples illustrate that our approach handles severe noise and ambiguity, which are qualities also present in Wikidata/DBpedia.
>
> That said, we agree with the reviewer that validating on dynamic KGs will further strengthen our claims. We have started adapting our framework to Wikidata, addressing engineering challenges. This process will take some time, but we will make every effort to include results in the revised or camera-ready version.
>
> ## Weakness 2
> Thank you for your valuable suggestion on grouping and standardizing comparisons by model family or parameter size.
> To address this, we conducted additional experiments on the CWQ dataset using Qwen2.5-instruct (7B / 32B / 72B) and Qwen3-32B models, so that performance scaling can be observed more clearly.
>
> | Model                  | Params | Complete Match  | LLM Call | Input Tokens | Output Tokens | Total Tokens |
> |------------------------|:------:|:--------------------:|:---------:|:-------------:|:--------------:|:-------------:|
> | Qwen2.5-instruct-7B    | 7B     | 59.6                |  9.50    |   3379.90    |     389.30    |    3769.20   |
> | Qwen2.5-instruct-32B   | 32B    | 61.0          |  9.19    |   3348.53    |     581.90    |    3930.43   |
> | Qwen3-32B              | 32B    | 61.9       |  9.80    |   3610.15    |     668.58    |    4338.73   |
> | Qwen2.5-instruct-72B   | 72B    | 63.9      |  9.40    |   4644.70    |    1201.00    |    5845.70   |
>
> From the results, we can see that increasing model parameters leads to steady improvements in Complete Match scores, largely due to richer internal knowledge and stronger semantic reasoning capabilities. t is worth noting that, in the Qwen series, the Qwen3 model was trained with roughly twice the number of tokens compared to the Qwen2.5 model, which generally implies more comprehensive internal knowledge. Although Qwen3-32B also benefits from architectural and training improvements, its performance gain over Qwen2.5-instruct-32B is smaller than the gain in training tokens. This confirms that while our approach can leverage internal knowledge, it does not solely rely on it. Even in scenarios where the model’s internal knowledge is insufficient or hallucinatory, our method remains robust and capable of producing correct results.
>
> ## Weakness 3
> We appreciate the reviewer’s insightful comment regarding the potential risk of hallucination or alignment drift when relying on LLM responses. We have explicitly considered this problem and adopted multiple, integrated components to minimize its impact:
>
> 1. **Conservative use of LLMs**
>    - LLMs are used only for **starting point discovery**, **probe generation**, and **semantic consistency filtering**.
>    - The filtering task is formulated as a *multiple-choice problem*, not free-form generation, to inherently reduce hallucinations.
>
> 2. **Systematic Verification Mechanisms**
>    - **No direct use of raw LLM outputs** in KG construction.
>    - **Probe Verification** (*Sec. 4.2*): Ensures only credible probes advance.
>    - **Dual Checks for Entity Addition** (*Sec. 4.3*): Untrusted probes are subject to *existence* and *necessity* checks before acceptance.
>    - **Structural Disambiguation** (*Sec. 4.4*): The *Credible Knowledge Disambiguation* step filters out relations structurally inconsistent with the KG, catching residual errors.
>
> 3. **Empirical Evidence**
>    - **Case Study #2** (in our public OpenReview reply) illustrates this empirically — a hallucinated relation was successfully eliminated by the structural disambiguation step.
>
> **Summary:**
> Our framework incorporates multiple, specific designs — including verification checks, constrained LLM tasks, and structural filtering — that work in concert to minimize the impact of LLM hallucinations and alignment drift.

---

> > ### Author Response · Authors · 2025-11-20
> > **Response to Reviewer 9oaZ  (2/2)**
> >
> > ## Weakness 4
> > We thank the reviewer for the valuable comment. In fact, Table 2 (Section 5.3) reports the main computational cost comparison between our method and baselines, including the number of LLM calls and token consumption, which are the dominant cost factors in all methods. We did not include wall-clock time because our experiments rely on API-based LLM usage, where variable network latency and occasional service throttling can make time measurements unfair and irreproducible. The reported metrics are stable and allow a reasonable estimation of runtime.

---

> ### Comment · Reviewer_9oaZ · 2025-11-27
>
> Thank you for your reply! All other concerns have been addressed, except for the first one. I believe that running the proposed scheme on a large knowledge graph with noise for comparison would be a particularly important experiment.

---

> > ### Author Response · Authors · 2025-12-02
> > **Response to Reviewer 9oaZ**
> >
> > Thank you for highlighting this important point. We fully agree that evaluating the proposed scheme on a larger and noisier knowledge graph is crucial for demonstrating its robustness. Before the rebuttal deadline, we conducted additional experiments on **Wikidata**, and the detailed results and analysis are presented in the *Additional Experiments on Wikidata*.
> >
> > Compared to Freebase, Wikidata has a more complex structure and contains many highly similar entities, making it a more challenging and realistic testbed. The experimental results show that our method maintains strong performance under this setting, further validating its generality and robustness.
> >
> > In addition, due to occasional 403 Forbidden responses when accessing Wikidata through its official API, the experiments encountered extra data incompleteness. Nevertheless, even under these more adverse conditions, our bridging mechanism effectively compensated for missing KG information and ensured strong overall performance. This behavior is fully aligned with the core design principle of our method:
> >
> > “Real-world KGs are inherently incomplete. A truly robust framework must not only leverage the KG’s verifiability but also compensate for its incompleteness.”

---

### Author Response · Authors · 2025-11-20
**Case Study**

We greatly appreciate the time and thoughtful feedback from all reviewers. To better illustrate the behavior and effectiveness of our method, we provide two additional case studies below.

## Case Study 1
| Steps | Content |
| :--- | :--- |
| **Question** | What country bordering France contains an airport that serves Nijmegen? |
| **Guidance Graph Construction** | **Guidance Graph:** `(country, borders, France)`, `(country, contains, airport)`, `(airport, serves, Nijmegen)`**Specific entities:** France; Nijmegen **Note:** France is retrievable in the KG and is used as the starting point. |
| **First Round** | **Baseline Method:** Retrieves all relations associated with France (498 in total), randomly selects a subset, and uses an LLM for pruning. **Our Method:** **Triplet to parse:** `(country, borders, France)` **Probe generation:** `[Germany, Spain, Italy, Belgium, Switzerland]` **KG verification & connectivity check:** France is connected to the probes via the path `location.location.adjoin_s → location.adjoining_relationship.adjoins`. **Candidate acquisition:** All nodes connected by the same relation path are collected as country candidates: `['Germany', 'United Kingdom', 'Italy', ..., 'Bay of Biscay', 'Dover']` (19 nodes in total). |
| **Second Round** | **Baseline Method:** Retrieves relations associated with the previous round's nodes, randomly selects a subset, and uses an LLM for pruning. **Our Method:** **Triplet to parse:** `(country, contains, airport)` **Probe generation:** `[Heathrow Airport, Charles de Gaulle Airport, Frankfurt Airport, Schiphol Airport, Munich Airport]` **KG verification & connectivity check:** The nodes from the previous round are connected to the probes via `location.location.contains`. **Candidate acquisition:** 15,184 nodes connected by the same relation path are collected as airport candidates. |
| **Third Round** | **Baseline Method:** Retrieves relations associated with the previous round's nodes and uses an LLM for pruning. **Our Method:** **Triplet to parse:** `(airport, serves, Nijmegen)` **Credible Knowledge Disambiguation:** Triggered because Nijmegen is a specific entity. **Disambiguation process:** Among the 15,184 candidate nodes from the previous round, only Weeze Airport is connected to Nijmegen via `aviation.airport.serves`. All other unconnected nodes are discarded. |
| **Knowledge Path** | `('France', 'location.location.adjoin_s→location.adjoining_relationship.adjoins', 'Germany')``('Germany', 'location.location.contains', 'Weeze Airport')``('Weeze Airport', 'aviation.airport.serves', 'Nijmegen')` |
| **Answer** | Germany |

In this case, the underlying subgraph is extremely dense and noisy. The first reasoning step involves 498 distinct relations from *France*—making full pruning computationally prohibitive, while random sampling (as used by baselines) risks missing the correct relation entirely.

Our method circumvents this by leveraging the LLM’s internal knowledge to generate targeted probes (e.g., neighboring countries like *Germany*, *Belgium*), which rapidly identify the unique relevant relation path (`location.location.adjoin_s → ...`) without exhaustive search. Since only one valid relation emerges, semantic consistency filtering is unnecessary.

Similarly, in the second round, probe-based retrieval efficiently isolates the correct relation linking candidate countries to airports. Finally, our **Credible Knowledge Disambiguation** mechanism exploits KG connectivity to filter 15,184 airport candidates down to a single node—*Weeze Airport*—that actually serves *Nijmegen*. Among all country candidates from Round 1, only *Germany* connects to *Weeze Airport*, yielding the correct answer.

---

> ### Author Response · Authors · 2025-11-20
> **Case Study 2**
>
> ## Case Study 2
> | Steps | Content |
> |---|---|
> | **Question** | Where did the "Country Nation World Tour" concert artist go to college? |
> | **GUIDANCE GRAPH CONSTRUCTION** | **Guidance Graph**: (concert artist, of, country nation world tour), (concert artist, attended, college) **Specific entities**: country nation world tour **Note:** country nation world tour is retrievable in the KG and serves as the starting point. |
> | **First Round** | **Triple to parse**: (concert artist, of, country nation world tour) **Probe Generation**: [Taylor Swift, Luke Bryan, Carrie Underwood, Blake Shelton, Keith Urban] **Verify probes in KG and check connectivity to the starting point**: country nation world tour has no connecting edge to the probes. **Semantic consistency filtering**: Among all 17 relations, `music.concert_tour.artist` is semantically consistent with the triple to parse. **Get all nodes connected via the same relation as the candidate set for concert artist**: ['Brad Paisley'] **Verified probes also serve as the candidate set for concert artist**: ['Taylor Swift', 'Luke Bryan', 'Carrie Underwood', 'Blake Shelton', 'Keith Urban'] |
> | **Second Round** | **Triple to parse**: (concert artist, attended, college) **Probe Generation**: [Belmont University, University of North Carolina at Chapel Hill, Oklahoma State University, Lipscomb University, University of Central Oklahoma] **Verify probes in KG and check connectivity to nodes from the previous round**: Only Brad Paisley connects to the probe Belmont University via `people.person.education→education.education.institution`. **Get 3 nodes from the same relation path as college candidates**: ['Belmont University', 'West Liberty University', 'John Marshall High School'] |
> | **Knowledge Path** | ('Brad Paisley', 'people.person.education→education.education.institution', 'Belmont University'), ('Country Nation World Tour', 'music.concert_tour.artist', 'Brad Paisley'), ('Brad Paisley', 'people.person.education→education.education.institution', 'West Liberty University'), ('Brad Paisley', 'people.person.education→education.education.institution', 'John Marshall High School') |
> | **Answer** | Belmont University, West Liberty University |
>
> The previous case demonstrated the strong capability of our method when combining knowledge probes with the KG structure.  The second case illustrates the robustness of our approach when the generated probes are hallucinations.
>
> In this case, the probes generated in the first round had no actual connections to the starting node *Country Nation World Tour* in the KG (in fact, these probes were all hallucinations).  However, our **semantic consistency filtering** was still able to select the semantically correct relation `music.concert_tour.artist` from the 17 outgoing relations of *Country Nation World Tour*, and identify the node **Brad Paisley**, which actually exists in the KG.  Considering that “missing edges” may occur in the KG, we retained unconnected probes in the first round as “temporary bridge” candidates.
>
> In the second round, among the generated probes, only **Belmont University** had a real connection to Brad Paisley in the KG via the relation  `people.person.education → education.education.institution`;  all other probes in this round were again hallucinations.
> By leveraging this single correct probe, we quickly locked onto the correct relation path and obtained other actual results (e.g., **West Liberty University**).
> The temporary bridge nodes preserved from the first round lacked correct connections in the second round and were therefore removed completely.

---

### Author Response · Authors · 2025-12-02
**Additional Experiments on Wikidata**

Due to the scale and dependency complexity of Wikidata, fully deploying it locally within the rebuttal period was not feasible. As an alternative, we follow the reviewers’ request by accessing Wikidata through its official API. The API occasionally triggers 403 Forbidden responses, which introduces a certain level of data incompleteness. We report results under this constrained setting below.

| Method | CWQ | QALD10-en |
|-------|:---:|:----------:|
| **ToG** | | |
| w/ Freebase | 58.8 | - |
| w/ Wikidata | 54.9 | 50.2 |
| **Performance Drop** | 6.6% | - |
| **ToG-R** | | |
| w/ Freebase | 59.2 | - |
| w/ Wikidata | 51.9 | 48.6 |
| **Performance Drop** | 12.3% | - |
| **Ours** | | |
| w/ Freebase | 75.2 | - |
| w/ Wikidata | 65.4 | 70.7 |
| **Performance Drop** | 13.0% | - |

We compare against **ToG** and **ToG-R**, which are among the few existing methods evaluated on Wikidata. ToG results are based on ChatGPT as reported in their paper, while our method uses DeepSeek-V3.
(**ToG reference**: *Think-on-Graph: Deep and Responsible Reasoning of Large Language Model on Knowledge Graph*. ICLR 2024.)

### Results on CWQ with Wikidata
CWQ is built on **Freebase**, so switching to Wikidata naturally introduces schema and entity-mapping mismatches. Consequently, all methods (ToG, ToG-R, and ours) exhibit performance drops, which is expected. Our drop ratio is comparable to ToG-R, and importantly, our absolute performance remains strong even under this cross-KG setting.

Because the Wikidata API occasionally returns incomplete results, the proportion of questions involving **at least one trusted bridging edge** increases from **21.8% to 30.9%**. This reflects two factors:

- inherent mismatch between CWQ and Wikidata
- additional incompleteness introduced by unavailable API edges

Even under these harsher conditions, our method remains robust—exactly aligning with our design principle:

> “Real-world KGs are inherently incomplete. A truly robust framework must not only leverage the KG’s verifiability but also compensate for its incompleteness.”

### Results on QALD10-en
Within the rebuttal time limit, we additionally evaluate on **QALD10-en**, a dataset natively built on Wikidata. Our method again achieves strong results, demonstrating its generality and effectiveness on real Wikidata-based QA tasks.

---

### Note · Program_Chairs · 2026-01-17
**Submission Desk Rejected by Program Chairs**

The following references in this submission do not refer to real documents and/or have major errors in bibliographic information:

 Sandeep Sen et al. Knowledge graph question answering: A benchmark and model. In Proceedings of the 2021 Conference on Empirical Methods in Natural Language Processing (EMNLP), 2021.
Jacob Cohen et al. Knowledge graph question answering: A survey. In Proceedings of the 2023 Annual Meeting of the Association for Computational Linguistics (ACL), 2023.